# Maximum Class Separation as Inductive Bias in One Matrix

**Tejaswi Kasarla**
University of Amsterdam

**Gertjan J. Burghouts**
TNO, Intelligent Imaging

**Max van Spengler**
University of Amsterdam

**Elise van der Pol**
Microsoft Research AI4Science

**Rita Cucchiara**
University of Modena
and Reggio Emilia

**Pascal Mettes**
University of Amsterdam

## Abstract

Maximizing the separation between classes constitutes a well-known inductive bias in machine learning and a pillar of many traditional algorithms. By default, deep networks are not equipped with this inductive bias and therefore many alternative solutions have been proposed through differential optimization. Current approaches tend to optimize classification and separation jointly: aligning inputs with class vectors and separating class vectors angularly. This paper proposes a simple alternative: encoding maximum separation as an inductive bias in the network by adding one fixed matrix multiplication before computing the softmax activations. The main observation behind our approach is that separation does not require optimization but can be solved in closed-form prior to training and plugged into a network. We outline a recursive approach to obtain the matrix consisting of maximally separable vectors for any number of classes, which can be added with negligible engineering effort and computational overhead. Despite its simple nature, this one matrix multiplication provides real impact. We show that our proposal directly boosts classification, long-tailed recognition, out-of-distribution detection, and open-set recognition, from CIFAR to ImageNet. We find empirically that maximum separation works best as a fixed bias; making the matrix learnable adds nothing to the performance. The closed-form matrices and code to reproduce the experiments are available on github.[1]

## 1 Introduction

An inductive bias of a learning algorithm describes a set of assumptions about the target function independent of training data [37]. Inductive biases play a vital role in the design of machine learning algorithms, consider for example inductive biases for image structures (*e.g.,* the convolution operator), symmetries (*e.g.,* rotational equivariance), or relational structures (*e.g.,* graph layers). Specifically for categorization, an important and long-standing inductive bias is optimal class separation. Perhaps the most well-known example of the use of this bias is Support Vector Machines, which maximize the margin of the hyperplane between samples of two classes [6]. Given many possible hyperplanes, the rationale is to select the one that represents the largest margin, *i.e.,* separation, known as a maximum-margin classifier. Similarly, boosting algorithms give weights to samples depending on their margin, such as AdaBoost which focuses on low margin samples [14]. In such algorithms and in many more, separation has been incorporated in their design to pull apart the classes, thereby improving generalization in classification settings to unseen data.

---

[1]https://github.com/tkasarla/max-separation-as-inductive-bias

36th Conference on Neural Information Processing Systems (NeurIPS 2022).

In deep learning-based classification, neural networks are by default not equipped with a maximum separation inductive bias. The cross-entropy loss with a softmax function is a common choice to train a classifier, aiming for discriminative power. Yet this framework does not explicitly drive maximum separation between the classes. Liu *et al.* [31] observed that networks by themselves try to decouple inter-class differences based on angles and intra-class variations based on norms. This decoupling is however limited out-of-the-box and a wide range of works have investigated optimization strategies to explicitly force classes away from each other. A common approach is to introduce additional losses which incorporate a notion of class margins, see *e.g.,* [25, 29, 33, 34] or by including orthogonality constraints [25]. Specifically, hyperspherical uniformity has shown to be an effective way to separate classes, whether it be as additional objectives in a softmax cross-entropy optimization [26, 29, 30, 70] or through hyperspherical prototypes [36].

This paper provides an alternative view to embed separation in deep networks with a straightforward solution: encoding maximum separation as an inductive bias in the network architecture itself with one fixed matrix. Rather than treating separation as an optimization problem, we find that the problem can be optimally solved in closed-form. The closed-form solution is given by a recursive algorithm which embeds all class vectors uniformly, such that all are equiangular and with zero mean. The solution holds for any finite number of classes and results in a single matrix that can be added *a priori* as a matrix multiplication on top of any network. The outputs of the proposed embedding are the inner product scores that feed into the softmax function, allowing us to still leverage powerful objectives such as cross-entropy on softmax activations.

We demonstrate that remarkably, adding only one fixed matrix in the neural network provides compelling improvements across various classification tasks. We show that enriching ResNet with a maximum separation inductive bias improves standard classification and especially long-tailed recognition. We also show that our proposal improves methods specifically designed for long-tailed recognition, highlighting its general utility. The results generalize to ImageNet and we show improvement in both shallow and deep networks. The added inductive bias helps to balance classes by design, avoiding the under-representation of rare classes. Beyond classification, we illustrate that out-of-distribution and open-set recognition benefit from an embedded maximum separation. Our general-purpose solution comes with no measurable difference in runtime and can be added in one line of code, enabling it to be adopted generally. To highlight the broad applicability of our proposal, we also show how task-specific state-of-the-art approaches for open-set recognition benefit directly when adding the proposed maximum separation.

## 2 Closed-Form Maximum Separation between Classes

Classification constitutes a supervised setting, where we are given a training set of $N$ examples $\{(\mathbf{x}_i, y_i)\}_{i=1}^N$, where $\mathbf{x}_i$ denotes input data at index $i$ and $y_i \in \{1, \ldots, C\}$ denotes the corresponding label for one of $C$ classes. Conventionally, this problem can be addressed by feeding inputs through a network, *i.e.,* $\mathbf{o}_i = \Phi(\mathbf{x}_i) \in \mathbb{R}^C$. The outputs denote class logits, which serve as input for a softmax activation followed by a cross-entropy loss.

Our goal is to embed maximum separation as an inductive bias into any network, so that the network no longer needs to learn this behaviour. We do so by adding a single matrix multiplication on top of the output of network $\Phi'(\cdot)$. The matrix itself should contain all necessary information about maximum separation between classes. While the general problem of optimally separating arbitrary numbers of classes on arbitrary numbers of dimensions is an open problem [29, 36], we note that we only need maximum separation of $C$ classes in a particular output space. To that end, let $\hat{\mathbf{x}}_i = \Phi'(\mathbf{x}_i) \in \mathbb{R}^{C-1}$ now denote the output of a network to a $(C-1)$-dimensional output space. We add the matrix multiplication to obtain $C$-dimensional class logits: $\mathbf{o}_i = P^T \hat{\mathbf{x}}_i$ with $P \in \mathbb{R}^{(C-1) \times C}$. The matrix $P$ denotes a set of $C$ vectors, one for each class, each of $(C-1)$ dimensions. This matrix is fixed such that the class vectors are separated both uniformly and maximally with respect to their pair-wise angles, which we in turn embed on top of a network. Below, we outline the definition and implementation of class vectors on a hypersphere with maximum separation. In what follows, the number of classes is denoted by $k + 1 = C$ for notational convenience. Moreover, this definition is restricted to matrices of the form $\mathbb{R}^{k \times (k+1)}$.

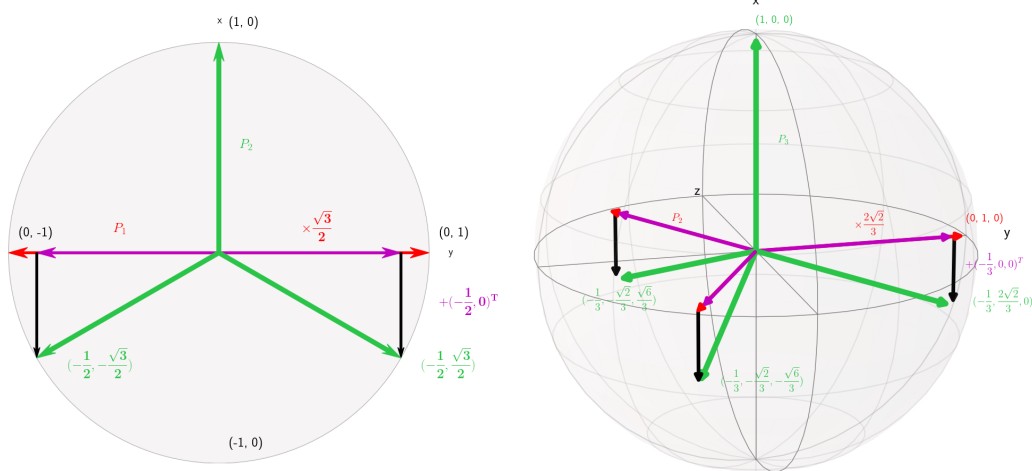

(a) Recursive update from 2 to 3 classes.  (b) Recursive update from 3 to 4 classes.

Figure 1: **Closed-form separation through recursion.** In (a), we visualize how to obtain maximally separated vectors for three classes on $\mathbb{S}^1$ given two class vectors on the line. In (b), we do the same for the step from three to four classes on $\mathbb{S}^2$. For any new $(k+1)$-th class (red), we introduce a new $k$-th dimension and position the new prototype along its axis $(1, 0, \ldots, 0)$. All other prototypes are shrunken by a factor $\sqrt{1 - \frac{1}{k^2}}$ (purple) and repositioned such that their center is placed at $(-\frac{1}{k}, 0, \ldots, 0)$ in the hyperplane perpendicular to the new class (green). Now, the matrices $P_2$ and $P_3$ are the matrices containing the green vectors shown in (a) and (b), respectively, as their columns. This recursive approach ensures that all classes remain of unit norm, with equal pair-wise similarity, and with zero mean.

**Definition 1.** (Maximally separated matrix) *For $k+1$ classes, let $P_k = [p_0, \ldots, p_k]$ denote a matrix of $k+1$ column vectors $\{p_i\}_{i=0}^{k} \subset \mathbb{S}^{k-1}$, such that $\forall_{i,j,k,i\neq j,j\neq k} \langle p_i, p_j \rangle = \langle p_i, p_k \rangle$ and $\sum_{i=0}^{k} p_i = 0$.*

Here, $\langle \cdot, \cdot \rangle$ denotes the cosine similarity. Definition 1 ensures that all class vectors are as far away from each other as possible based on two requirements. First, the angle between any two class vectors should be the same. Equal pair-wise similarity entails uniformity as any deviation from this setup leads to classes that move closer to each other [50]. Second, the mean over all class vectors should be the origin. This constraint prevents trivial solutions such as an identical embedding of all classes as well as suboptimal embeddings in terms of separation such as one-hot encodings. In short, the definition prescribes that $k+1$ classes can be optimally separated specifically when using $k$ output dimensions. To be able to construct such an embedding, we first need to know what the pair-wise angular similarity is for any fixed number of classes:

**Lemma 1.** *For maximally separated matrix $P_k$, $\forall_{i,j,i\neq j} \langle p_i, p_j \rangle = -\frac{1}{k}$.*

*Proof.* With $P_k$ maximally separated, we have $\sum_{i=0}^{k} p_i = 0$ and hence:

$$0 = \langle p_0, \sum_{i=0}^{k} p_i \rangle = 1 + \sum_{i=1}^{k} \langle p_0, p_i \rangle = 1 + k \langle p_0, p_1 \rangle, \tag{1}$$

since all class vectors are at equal angle to each other. We have $\langle p_0, p_1 \rangle = -\frac{1}{k}$ and again by extension of equidistance: $\forall_{i,j,i\neq j} \langle p_i, p_j \rangle = -\frac{1}{k}$. ☐

The implication of the lemma above is that an optimal separation of $k+1$ classes on the manifold $\mathbb{S}^{k-1}$ results in classes being separated beyond orthogonality, a direct result of using the full space and different from classical one-hot encodings, which only span the positive sub-space and

therefore provide only orthogonal separation. The outcome of the lemma allows us to construct $P_k$ recursively [1, 38].

For any finite number of classes $k + 1$, the closed-form solution for $P_k$ is given as:

$$P_1 = \begin{pmatrix} 1 & -1 \end{pmatrix} \in \mathbb{R}^{1 \times 2} \tag{2}$$

$$P_k = \begin{pmatrix} 1 & -\frac{1}{k}\mathbf{1}^T \\ \mathbf{0} & \sqrt{1 - \frac{1}{k^2}} P_{k-1} \end{pmatrix} \in \mathbb{R}^{k \times (k+1)} \tag{3}$$

The closed-form solution for two classes ($P_1$) is trivially given as +1 and -1 on the line. For more than 2 classes, $k \geq 2$, we start from the solution given for $k - 1$ on $\mathbb{S}^{k-2}$. We add the vector for the new class as $p_1 = (1, 0, \ldots, 0)$. We place the solution $P_{k-1}$ at $(-\frac{1}{k}, 0, \ldots, 0)$, in the hyperplane perpendicular to $p_k$, where $P_{k-1}$ is shrunk so that its radius is $\sqrt{1 - \frac{1}{k^2}}$. In this manner, all points in $P_k$ are of radius 1 and uniformly distributed, as proven below. Figure 1 shows the construction of our embeddings for 3 and 4 classes.

**Theorem 1.** *For any $k \geq 1$, $P_k$ is a maximally separated matrix.*

*Proof.* We proceed by induction. Note that, for $k = 1$, the angular similarity of the two vectors is clearly given by $-\frac{1}{k} = -1$. Now, assume the theorem holds for $k - 1 \in \mathbb{N}$ and let $(P_n)_{*i}$ denote the $i$-th column vector of $P_n$ for arbitrary $n \in \mathbb{N}$. Then, for $k$ and $i \neq j$, we obtain the case where either $i$ or $j$ is 1 and the case where neither is 1. For the first case, without loss of generality assume that the $i$-th element is the first, so that:

$$\langle (P_k)_{*i}, (P_k)_{*j} \rangle = -\frac{1}{k} + \sqrt{1 - \frac{1}{k^2}} \langle \mathbf{0}, (P_{k-1})_{*j} \rangle = -\frac{1}{k}.$$

For the second case, we derive:

$$\begin{aligned} \langle (P_k)_{*i}, (P_k)_{*j} \rangle &= \frac{1}{k^2} + \left(1 - \frac{1}{k^2}\right) \langle (P_{k-1})_{*i}, (P_{k-1})_{*j} \rangle \\ &= \frac{1}{k^2} - \left(1 - \frac{1}{k^2}\right)\left(\frac{1}{k-1}\right) \\ &= -\frac{1}{k}, \end{aligned}$$

since $\langle (P_{k-1})_{*i}, (P_{k-1})_{*j} \rangle = -\frac{1}{k-1}$ by assumption. Therefore, the vectors of $P_k$ have a pair-wise angular similarity of $-\frac{1}{k}$. To prove that $P_k$ has zero mean, we again proceed by induction. Note that for $P_1$, the statement obviously holds. Now, assume the statement holds for $k - 1 \in \mathbb{N}$, then

$$\sum_{i=0}^{k} (P_k)_{*i} = \begin{pmatrix} 1 - k * \frac{1}{k} \\ \sqrt{1 - \frac{1}{k^2}} \sum_{i=0}^{k-1} (P_{k-1})_{*i} \end{pmatrix} = \mathbf{0},$$

since $\sum_{i=0}^{k-1} (P_{k-1})_{*i} = \mathbf{0}$ by assumption. $\square$

For $C = k + 1$ classes, the solution $P_k$ consists of $C$ vectors of unit length. The main idea of this work is to utilize $P_k$ to obtain class logits under a fixed maximum separation, see Figure 2. For sample $\mathbf{x}_i$, we want its network output $\hat{\mathbf{x}}_i = \Phi'(\mathbf{x}_i)$ to align with the vector of its corresponding class. Given $P_k$, the alignment is maximized if the network output points in the same direction as the class vector. This can be obtained by computing the dot product between both. Since we need to compute the logits for all class vectors in the final softmax activation, we simply

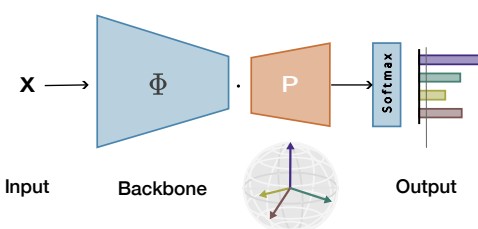

Figure 2: **Maximum separation pipeline.**

augment the network output with a single matrix multiplication: $\mathbf{o}_i = P^T \hat{\mathbf{x}}_i$. By enforcing maximum separation from the start, a network no longer needs to learn this behaviour, essentially simplifying the learning objective to maximizing the alignment between samples and their fixed class vectors.

| | CIFAR-100 | | | | | CIFAR-10 | | | | |
|---|---|---|---|---|---|---|---|---|---|---|
| | - | 0.2 | 0.1 | 0.02 | 0.01 | - | 0.2 | 0.1 | 0.02 | 0.01 |
| ConvNet | 56.70 | 45.97 | 40.34 | 27.35 | 16.59 | 86.68 | 79.47 | 73.90 | 51.40 | 43.67 |
| + This paper | **57.05** | **46.59** | **40.44** | **28.27** | **18.40** | **86.76** | **79.63** | **75.88** | **55.25** | **48.05** |
| | +0.35 | +0.62 | +0.10 | +0.92 | +1.81 | +0.08 | +0.16 | +1.98 | +3.85 | +4.38 |
| ResNet-32 | 75.77 | 65.74 | 58.98 | 42.71 | 35.02 | 94.63 | 88.17 | 83.10 | 68.64 | 56.98 |
| + This paper | **76.54** | **66.01** | **60.54** | **45.12** | **38.85** | **95.09** | **91.42** | **88.16** | **77.02** | **69.70** |
| | +0.77 | +0.27 | +1.56 | +2.41 | +3.83 | +0.46 | +3.25 | +5.06 | +8.38 | +12.72 |

Table 1: **Adding maximum separation as inductive bias on CIFAR-10 and CIFAR-100** for standard and imbalanced settings. Both the AlexNet and a ResNet architectures benefit from an embedded maximum separation, especially when imbalance increases and networks are more expressive.

We note that the norm of the class vectors is not taken into account during their construction as we operate on the hypersphere. We can optionally include a radius hyperparameter which controls the norm of the class vectors, *i.e.*, the network output is given as: $\mathbf{o}_i = \rho P^T \hat{\mathbf{x}}_i$ with $\rho$ as hyperparameter. In practice however, the radius of the class vectors is of little influence to the task performance and we set it to 1 unless specified otherwise.

# 3 Experiments

**Implementation details.** Across our experiments, we train a range of network architectures including AlexNet [22], multiple ResNet architectures [17, 67], and VGG32 [47], as provided by PyTorch [40]. All networks are optimized with SGD with a cosine annealing learning rate scheduler, initial learning rate of 0.1, momentum 0.9, and weight decay 5e-4. For AlexNet we use a step learning rate decay scheduler. All hyperparameters and experimental details are also available in the provided code.

## 3.1 Classification and long-tailed recognition

**Classification with maximum separation.** In the first set of experiments, we evaluate the potential of maximum separation as inductive bias in classification and long-tailed recognition settings using the CIFAR-100 and CIFAR-10 datasets along with their long-tailed variants [11]. Our maximum separation is expected to improve learning especially when dealing with under-represented classes which will be separated by design with our proposal. We evaluate on a standard ConvNet and a ResNet-32 architecture with four imbalance factors: 0.2, 0.1, 0.02, and 0.01. We set $\rho = 0.1$ for ResNet-32 as this provides a minor improvement over $\rho = 1$. The results are shown in Table 1. We report the confidence intervals for these experiments in the supplementary materials. For an AlexNet-style ConvNet, classification accuracy improves in all settings, especially when imbalances is highest. With a more expressive network such as a ResNet, we find that embedding maximum separation provides a bigger boost in performance. The higher the imbalance, the bigger the improvements; on CIFAR-100 the accuracy improves from 35.02 to 38.85, on CIFAR-10 from 56.98 to 69.70, an improvement of 12.72 percent point. For further analysis, we have investigated the relation between accuracy improvement and train sample frequency. In Figure 3 we show the results over the imbalance factors, highlighting that our proposal improves all classes as imbalance increases, especially classes with lower sample frequencies.

In the supplementary materials, we furthermore report the Angular Fisher Score as given by Liu *et al.* [32]. This score evaluates the angular feature discriminativeness of the trained models. We also compare our closed-form fixed matrix to the optimization-based fixed matrix of Mettes *et al.* [36]. These results show that the closed-form maximum separation is more discriminative than optimization-based separation.

**Should maximum separation be a fixed inductive bias?** Separation in this work takes the form of a fixed matrix multiplication that can be plugged on top of any network. But is the information in this matrix as inductive bias optimal for classification? To test this hypothesis, we compare the previous results on CIFAR-100 and CIFAR-10 and their long-tailed variants [11] to two baselines. The first baseline makes the matrix $P$ learnable and our proposal acts as an initialization of the learnable matrix. Intuitively, if the results improve by making the matrix learnable instead of fixed, there is more information that is not captured in the closed-form solution. The second baseline

adds a learnable fully-connected layer with standard random initialization instead of a fixed matrix multiplication, to investigate whether our obtained improvements are not simply a result of extra computational efforts.

The results are shown in Figure 4. On both datasets across all imbalance factors, embedding maximum separation as a fixed constraint in a network works best. Making the matrix learnable actually decreases the performance, indicating that further optimization discards important information and highlighting that separation is an inductive bias we should not steer away from. Simply adding an extra learnable layer with standard initialization is not effective, which indicates that improvements are not guaranteed when adding more fully-connected layers. We conclude that maximum separation is best added in a rigid manner as strict constraint for networks to adapt to.

**Enriching long-tailed recognition approaches.** So far, we have shown that conventional network architectures are better off with maximum separation as inductive bias for classification and long-tailed recognition. In the third experiment, we investigate the potential of adding our proposal on top of methods that are specifically designed for long-tailed recognition. We perform experiments on two methods, LDAM [8] and MiSLAS [69]. LDAM addresses learning on imbalanced data through a label-distribution-aware margin loss which replaces the conventional cross-entropy loss. We can easily augment LDAM by placing the matrix on top of the used backbone. Their hyperparameter $\mu$ in LDAM is set to 0.5 when using their method as is and set to 0.4 when augmented with our proposal as this maximizes the scores for both settings. MiSLAS is a recent state-of-the-art approach for long-tailed recognition that improves two-stage methods by means of label-aware smoothing and shifted batch normalization. We again plug in our approach in the provided code simply as a fixed matrix multiplication on top of the used backbone.

In Table 2, we report the results on CIFAR-100 using a ResNet-32 backbone across three imbalance factors. For LDAM, we find that both for the SGD and the DRW variants, adding maximum separation on top improves the accuracy. For imbalance factor 0.01 for LDAM, we improve the accuracy from 39.87 to 42.02 with the SGD variant and from 42.37 to 43.19 for the DRW variant. The recent MiSLAS approach to long-tailed recognition also benefits from maximum separation after both stages. The most competitive results after their stage 2 are improved by 1.59, 0.83, and 0.42 percent point for respectively imbalance factors 0.1, 0.02, and 0.01. The long-tailed baselines are specifically designed to address class imbalance but additionally benefit from maximum separation across the board. We conclude that maximum separation as a fixed inductive bias complements task-specific approaches, highlighting its general applicability and usefulness.

**ImageNet experiments.** To highlight that our proposal is also beneficial with many classes and deeper networks, we provide results on ImageNet for two ResNet backbones in Table 3. With a ResNet-50 backbone, we obtain improvements of 1.6 percent point on ImageNet and 3.5 percent point on its long-tailed variant in top 1 accuracy. With a deeper ResNet-152, the improvements are 0.6 and 1.4 percent points for the standard and long-tailed benchmarks. We also find that the top 5 accuracy improves from 92.4 to 94.9 with ResNet-50 and from 94.3 to 95.1 with ResNet-152. We conclude that the proposed inductive bias in matrix form is also useful for larger-scale classification.

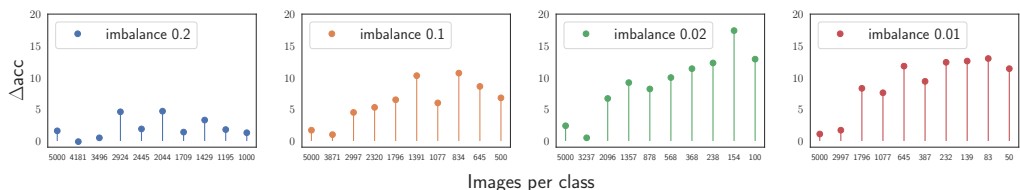

Figure 3: **Effect of maximum separation on class accuracy as a function of sample frequency.** We show the per class accuracy improvements when adding separation on CIFAR-10 with a ResNet-32 for imbalance factors (from left to right) 0.2, 0.1, 0.02, 0.01. The accuracy increases over all classes as imbalance increases, especially classes with lower sample frequencies.

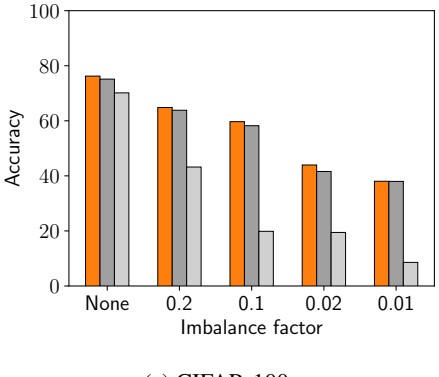

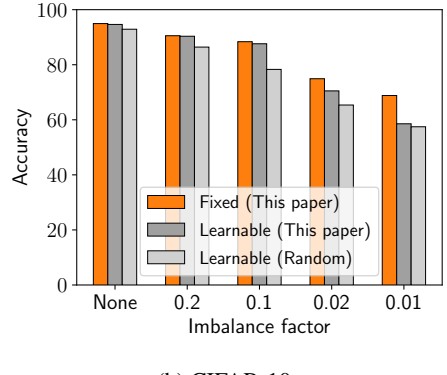

(a) CIFAR-100.                                                    (b) CIFAR-10.

Figure 4: **Should maximum separation be embedded as a fixed inductive bias?** On both CIFAR-100 and CIFAR-10, we find that embedding a fixed separation (orange) works best, outperforming baselines that seek to further optimize a matrix initialized with maximum separation and that simply add another learnable matrix initialized like any other layer. Maximum separation should be computed *a priori* and kept fixed in the network architecture.

|  | Imbalance factor | | |  |  | Imbalance factor | | |
|---|---|---|---|---|---|---|---|---|
|  | 0.1 | 0.02 | 0.01 |  |  | 0.1 | 0.02 | 0.01 |
| LDAM-SGD | 55.05 | 43.85 | 39.87 |  | MiSLAS (stage 1) | 58.36 | 44.69 | 40.29 |
| + This paper | **57.72** | **45.14** | **42.02** |  | + This paper | **59.63** | **45.65** | **40.56** |
|  | +2.67 | +1.29 | +2.20 |  |  | +1.27 | +0.96 | +0.27 |
| LDAM-DRW | 57.45 | 47.56 | 42.37 |  | MiSLAS (stage 2) | 61.93 | 52.53 | 48.00 |
| + This paper | **58.37** | **48.02** | **43.19** |  | + This paper | **63.52** | **53.36** | **48.42** |
|  | +0.92 | +0.46 | +0.82 |  |  | +1.59 | +0.83 | +0.42 |

Table 2: **Enriching imbalanced learning algorithms with our maximum separation** on CIFAR-100 for imbalance factors 0.1, 0.02, and 0.01. We perform experiments with two variants of LDAM of [8] and with the recent MiSLAS of [69] after both stages of their approach. All results are obtained by running the author-provided code. Across all methods and imbalances factors, we find that a maximum separation inductive bias provides a consistent boost in accuracy.

**On feature dimensionality and scaling.** We set the last layer feature dimension to $D = 512$ or 1024 depending on the dataset. The size of the learnable last layer is $D \times k$. On the top of this we add our fixed matrix of size $k \times k + 1$ to get class logits. This means that in terms of learnable parameters, there is no noticeable difference between the standard setup and our maximum separation formulation. With many classes however, the fixed final matrix will be large, which can lead to extra computational effort. At the ImageNet scale (1,000 classes) training and inference times are similar, but we have yet to investigate extreme classification cases.

### 3.2   Out-of-distribution detection and open-set recognition

We also investigate the potential of our proposal outside the closed set of known classes. To that end, we perform additional experiments on out-of-distribution detection and open-set recognition.

**Out-of-distribution with maximum separation.** Intuitively, maximum separation and uniformity between classes allows for more opportunities for samples outside the distribution of known classes to fall in between the spaces of class vectors, especially since all classes are positioned beyond orthogonal to each other as per Lemma 1. We evaluate this intuition first on out-of-distribution detection. Following Liu et.al. [28], we use CIFAR-100 as in-distribution set and experiment with SVHN and Placed 365 as out-of-distribution sets. Using the same experiment settings reported in their paper, we train on a WideResNet and compare the out-of-distribution performance on the common

| | Resnet-50 | | | | Resnet-152 | | | |
| --- | --- | --- | --- | --- | --- | --- | --- | --- |
| | top 1 | | top 5 | | top 1 | | top 5 | |
| | Base | + Ours | Base | + Ours | Base | + Ours | Base | + Ours |
| Imagenet | 73.2 | **74.8** | 92.4 | **94.9** | 77.9 | **78.5** | 94.3 | **95.1** |
| Imagenet-LT | 43.8 | **47.3** | 70.4 | **73.6** | 48.3 | **49.7** | 73.9 | **74.8** |

Table 3: **Evaluations on ImageNet and ImageNet-LT** for two ResNet architectures. Both the top 1 and top 5 accuracies improve on ImageNet when embedding maximum separation on top of the architectures, with a further boost for the long-tailed ImageNet variant[11].

| | Metric | w/o maximum separation | | | w/ maximum separation | | |
| --- | --- | --- | --- | --- | --- | --- | --- |
| | | FPR95 ↓ | AUROC ↑ | AUPR ↑ | FPR95 ↓ | AUROC ↑ | AUPR ↑ |
| SVHN | Softmax Score | 84.00 | 71.40 | 92.87 | **83.04** | **75.58** | **94.59** |
| | Energy Score | 85.76 | 73.94 | 93.91 | **78.86** | **85.42** | **96.92** |
| | Mahalanobis | 44.02 | 90.48 | 97.83 | **35.88** | **91.45** | **97.93** |
| Places 365 | Softmax Score | **82.86** | 73.46 | 93.14 | 83.08 | **73.53** | **93.54** |
| | Energy Score | **80.87** | 75.17 | 93.40 | 81.36 | **76.01** | **93.64** |
| | Mahalanobis | **88.83** | 67.87 | 90.71 | 89.16 | **69.33** | **91.49** |

Table 4: **Out-of-distribution detection on CIFAR-100** with and without maximum separation embedded in a Wide ResNet, using SVHN and Placed 365 as out-of-distribution datasets. On SVHN we observe improvements across all matrics and measures. On Places 365, out-of-distribution detection is better for the AUROC and AUPR metrics, but not the FPR95.

metrics FPR95, AUROC, and AUPR. We determine whether a sample is in- or out-of-distribution using three scoring functions. The first directly uses the maximum softmax probability to distinguish in- and out-of-distribution samples [19]. The second is the energy score as introduced in Liu *et al.* [28], which classifies an input as out-of-distribution if the negative energy score is smaller than a threshold value, The third is the confidence score based on Mahalanobis distance from Lee *et al.* [23].

The results are shown in Table 4. With SVHN as out-of-distribution set, embedding a maximum separation inductive bias improves all metrics across all out-of-distribution detection methods. Especially the energy score benefits from maximum separation with an improvement of 6.90 in FPR95, 11.48 in AUROC, and 3.01 in AUPR. On Places 365, the results are overall closer. Maximum separation improves on the AUROC and AUPR metrics but not the FPR95 metric. We observe that adding maximum separation performs best for near out-of-distribution data. In Figure 5 we provide further analyses on the effect of maximum separation for out-of-distribution detection with energy scores, highlighting that separation increases the gap in energy scores between both sets, making it easier to discriminate both. We conclude that a maximum separation inductive bias is valuable for out-of-distribution detection, especially for near out-of-distribution data.

**Open-set recognition with maximum separation.** We also investigate the potential of maximum separation for open-set recognition, which can be viewed as a generalization of out-of-distribution detection [35]. To take our proposal to the ultimate test, we start from the recent state-of-the-art work of Vaze *et al.* [55]. We experiment with adding our proposal to two approaches outlined in their work. The first applies various training tricks and hyperparameter tuning to improve the closed-set classifier, followed by open-set recognition using the maximum softmax probability (MSP+) score. The second performs the same training with the Maximum Logits Score (MLS) before the softmax activations to differentiate closed- from open-set examples. All experiments are run on author-provided code on four benchmarks: SVHN, CIFAR10, CIFAR + 10, and CIFAR + 50, where we use the 5 splits and report the average scores as outlined by Vaze *et al.* [55].

The results of the open-set experiments are shown in Table 5 with the AUROC metric. For both the MSP+ and the state-of-the-art MLS scores, maximum separation is a positive addition. This result follows the intuition of the original approach that improving the closed-set classifier – here by means of the extra inductive bias – is also beneficial for open-set recognition. We additionally visualize the 2*d* features of the last layer for our approach in the supplementary materials. The visualization shows

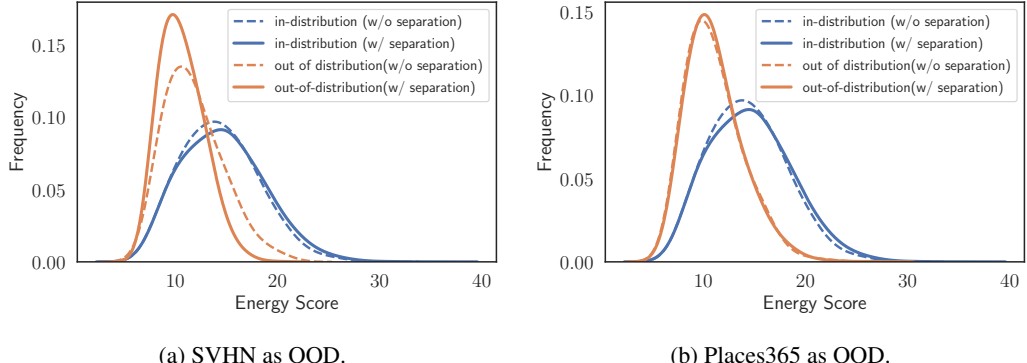

| | | |
|---|---|---|
| (a) SVHN as OOD. | | (b) Places365 as OOD. |

Figure 5: **Energy scores for out-of-distribution detection.** We plot the inverse of the energy scores for ease of visualization. With SVHN as out-of-distribution dataset for CIFAR-100 as in-distribution dataset, we find that embedding maximum separation increases the gap in energy scores between both datasets, making it easier to differentiate between both. With Places 365 as out-of-distribution dataset, the energy scores are affected less, resulting in smaller improvements with maximum separation.

| | SVHN | CIFAR10 | CIFAR + 10 | CIFAR + 50 |
|---|---|---|---|---|
| MSP+ | 95.94 | 90.10 | 94.48 | 93.58 |
| + This paper | **96.22** | **91.05** | **95.57** | **94.31** |
| | +0.28 | +0.95 | +1.09 | +0.73 |
| MLS | 97.10 | 93.82 | 97.94 | 96.48 |
| + This paper | **97.58** | **95.30** | **98.33** | **96.74** |
| | +0.48 | +1.48 | +0.39 | +0.26 |

Table 5: **Open-set recognition on four benchmarks** with and without maximum separation building on the experimental settings of Vaze et.al. [55]. Whether using the maximum probability or the recently proposed maximum logit to perform open-set recognition, a fixed maximum class separation helps to differentiate closed-set from open-set samples.

that, as training progresses, the norm of the known classes gets bigger than the norm of the open-set classes, making differentiation between both feasible.

# 4  Related work

In deep learning literature, the principle of separation has been investigated from several perspectives. Separation can be achieved by enforcing or regularizing orthogonality. For example, Li et al. [25] propose to regularise class vectors orthogonal to each other. Orthogonality is also often used as regularisation for intermediate network layers, see *e.g.,* [3, 42, 43, 59, 64, 65]. Orthogonal constraints are fixed and easy to obtain by design. For classification specifically however, orthogonality does not result in a maximal separation between classes, as only the positive sub-space is utilized. We strive for a solution that makes use of the entire output space.

Beyond orthogonality, multiple works improve class separation by promoting angular diversity between classes. A common approach to do so is by incorporating class margin losses [12, 32, 34, 58, 68]. For example, the generalized large-margin softmax loss enforces both intra-class compactness and inter-class separation [33]. These works are in line with the observations of Liu *et al.* [31] that deep networks have a natural tendency to break classification down into maximizing class separation and minimizing intra-class variance. In particular, separation has in recent years been successfully tackled by adopting the perspective of hyperspherical uniformity. Such uniformity can for example be optimized in deep networks by minimizing the hyperspherical energy between class vectors [26, 29, 30, 16]. Recently, Zhou *et al.* [70] extend this line of work by learning towards the largest margin with a zero-centroid regularization. Hyperspherical uniformity can also be approximated by minimizing the maximum cosine similarity between pairs of class vectors [15, 36, 61]. In this work,

we also start from hyperspherical uniformity and deviate from current literature on one important axis: hyperspherical uniformity does not require optimization. We show and prove that maximum separation has a closed-form solution and can be added as a fixed matrix to any network. Mroueh *et al.* [38] also show that such a closed-form solution is beneficial when added to SVM loss functions and mention it could be generalized to other loss functions. Having maximum separation as a fixed inductive bias does not require additional hyperparameters and comes with minimal engineering effort, lowering the barrier towards broad integration in the field.

Our work fits into a broader tendency of incorporating inductive biases into deep learning, for example by incorporating rotational symmetries [10, 13, 24, 46, 49, 51, 54, 57, 63], scale equivariance [48, 62], gauge equivariance [9, 18], graph structure [4, 7, 20, 56], physical inductive biases [2, 21, 41, 44], symmetries between reinforcement learning agents [5, 27, 53, 60], and visual inductive biases [45, 52, 66]. Embedding maximum separation as inductive bias is complementary to the listed works.

**Relation to Neural Collapse**   Neural Collapse is an empirical phenomenon that arises in the features of the last layer and the classifiers of neural networks during terminal phase training [39]. In neural collapse, the class means and the classifiers themselves collapse to the vertices of a simplex equiangular tight frame. From these insights, recent work by Zhu *et al.* [71] fix the last layer classifier with a fixed simplex of size $(k + 1) \times d$ with $d >= (k + 1)$ for $k + 1$ classes. Our work reaches a similar conclusion from the perspective of maximum separation and we find its potential also reaches long-tailed and out-of-distribution detection. Our recursive algorithm is more compact at size $(k + 1) \times k$ and we plug our approach on top of any network architecture, rather than replace the final layer as done in [71]. Due to the recursive nature of our approach, there is also potential for incremental learning with maximum separation.

## 5   Conclusions

This paper strives to embed a well-known inductive bias into deep networks: separate classes maximally. Separation is not an optimization problem and can be solved optimally in closed-form. We outline a recursive algorithm to compute maximum class separation and add it as a fixed matrix multiplication on top of any network. In this manner, maximum separation can be embedded with negligible effort and computational complexity. To showcase that our proposal is an effective building block for deep networks, we perform various experiments on classification, long-tailed recognition, out-of-distribution detection, and open-set recognition. We find that our solution to maximum separation as inductive bias (i) improves classification, especially when classes are imbalanced, (ii) is best treated as a fixed matrix, (iii) improves standard networks and task-specific state-of-the-art algorithms, and (iv) helps to detect samples from outside the training distribution. With separation decoupled and solved prior to training, a network only needs to optimize the alignment between samples and their fixed class vectors. We conclude that our one fixed matrix is an effective, broadly applicable, and easy-to-use addition to classification in networks. Our approach is currently focused on multi-class settings only, where exactly one label needs to be assigned to each example. Maximum separation does also not naturally generalize to zero-shot settings, as such settings require that classes are represented by semantic vectors that point in similar directions based on semantic similarities. While we focus on the classification supervised setting, maximum separation has potential for unsupervised learning as well as self-supervised learning paradigm. As the matrix imposes geometric structure in the output space it could potentially help in improving the uniformity of classes.

## Acknowledgements

This work is financially supported by the ELLIS Amsterdam Unit.

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
