# Maximum Class Separation as Inductive Bias in One Matrix

## Supplementary Material

## 1 Angular Fisher Score analysis

We report the Angular Fisher Score from Liu *et al.* [1] in the table below for CIFAR-10 and CIFAR-100 test sets. We trained a ResNet-32 with the same settings as Table-1 from the paper. For the Angular Fisher Score, lower is better. Across datasets and imbalance factors, the score is lower with maximum separation, providing additional verification of our approach.

| | CIFAR-10 | | | CIFAR-100 | | |
|---|---|---|---|---|---|---|
| | - | 0.1 | 0.01 | - | 0.1 | 0.01 |
| SCE | 0.0583 | 0.2305 | 0.4141 | 0.2954 | 0.4958 | 0.7202 |
| This Paper | **0.0555** | **0.1397** | **0.3240** | **0.1521** | **0.4483** | **0.6952** |

Table 1: **Angular Fisher Score** for standard and imbalanced settings. Lower fisher score indicates better discriminative features.

## 2 Comparison to optimization-based separation

We compare our approach to a baseline that optimizes for class vectors through optimization and fixes the vectors afterwards. One such methods is the hyperspherical prototype approach of Mettes *et al.* [2]. We have looked into the class vectors themselves, as well as the downstream performance. For the class vectors, we find that a gradient-based solution has a pair-wise angular variance of over one degree for 100 classes, indicating that not all classes are equally well separated, while we do not have such variability. We have also performed additional long-tailed recognition experiments for our maximum separation approach versus the hyperspherical prototype approach of Mettes *et al.* [2]. Below are the results for CIFAR-10 and CIFAR-100 for three imbalance ratios:

| | CIFAR-10 | | | CIFAR-100 | | |
|---|---|---|---|---|---|---|
| | - | 0.1 | 0.01 | - | 0.1 | 0.01 |
| Mettes *et al.* | 93.27 | 86.16 | 61.63 | 71.58 | 53.28 | 34.08 |
| This Paper | **95.09** | **88.16** | **69.70** | **76.23** | **60.54** | **38.85** |

Table 2: **Comparison to optimization approach** of Mettes *et al.* which first optimizates for maximally separated class vectors and fixes vectors during training.

We conclude that a closed-form maximum separation is preferred for recognition.

## 3 Error bars for Table 1

We have run the experiments in Table 1 of the main paper 5 times and added error bars. The results show that over multiple runs, the improvements are stable.

| | CIFAR-100 | | | | | CIFAR-10 | | | | |
|---|---|---|---|---|---|---|---|---|---|---|
| | - | 0.2 | 0.1 | 0.02 | 0.01 | - | 0.2 | 0.1 | 0.02 | 0.01 |
| ConvNet | 56.45± 0.32 | 45.88 ± 0.43 | 40.04± 0.38 | 27.17± 0.52 | 16.31 ± 0.22 | 86.30± 0.21 | 78.37± 1.04 | 73.6± 0.58 | 51.71 ± 0.38 | 42.72 ± 1.21 |
| + This Paper | **57.05± 0.55** | **46.21± 0.45** | **40.44± 0.23** | **28.16± 0.31** | **18.15 ± 0.53** | **86.48± 0.20** | **79.44± 1.20** | **75.4 ± 1.03** | **56.98 ± 1.16** | **48.26 ± 0.65** |
| ResNet-32 | 75.42± 0.37 | 65.20± 0.43 | 58.01± 1.01 | 42.70± 0.20 | 34.98± 0.54 | 94.41±0.25 | 87.96± 0.24 | 82.95± 0.45 | 68.04± 0.83 | 56.5± 0.56 |
| + This Paper | **76.41± 0.21** | **66.22± 0.56** | **60.23± 0.54** | **45.11± 0.13** | **37.65± 0.81** | **96.12± 0.19** | **91.26± 0.22** | **88.01± 0.73** | **77.12± 1.33** | **68.8± 1.42** |

Table 3: **Adding maximum separation as inductive bias on CIFAR-10 and CIFAR-100** for standard and imbalanced settings. Both the AlexNet and a ResNet architectures benefit from an embedded maximum separation, especially when imbalance increases and networks are more expressive.

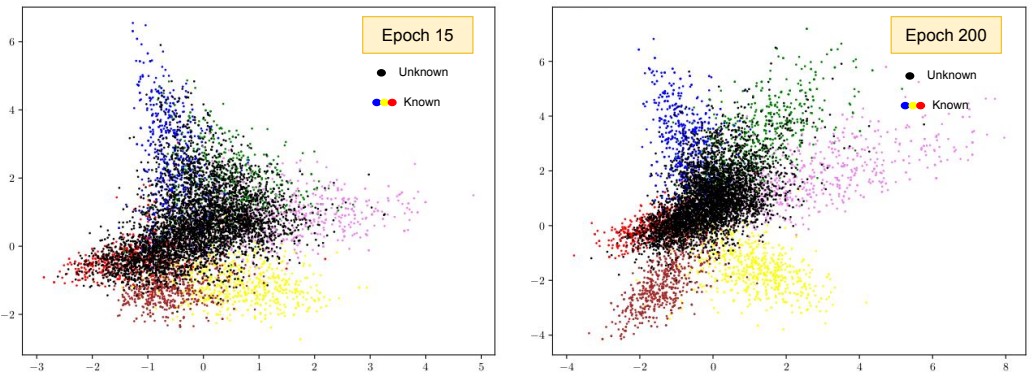

Figure 1: **OSR with maximum separation**

## 4 Analysis on open-set recognition

We follow analysis from appendix of Vaze *et al.* [3] and train the VGG-32 network for feature dimensions $D = 2$ for 200 epochs. We plot features at epochs 15 and 200 in Fig 1. As training progresses, the feature norm of unknown classes is gets smaller than known classes and maximum separation helps in maintaining both the class-wise separation and lower norm of unknown classes.