# OpenReview forum: "Maximum Class Separation as Inductive Bias in One Matrix"
_NeurIPS.cc/2022/Conference — NeurIPS 2022 Accept_

### Official Review · Reviewer_BhPp · 2022-06-28

**Rating:** 5
**Confidence:** 3
**Soundness:** 3 good
**Presentation:** 3 good
**Contribution:** 1 poor

**Summary:**

In this paper, the author explores how to achieve maximizing class separation by directly adding a pre-calculated layer into the deep neural networks. The weights of the layer are solved in closed-form by a recurve algorithm. The effectiveness of the maximally separable vectors is verified on various datasets from long-tailed recognition, out-of-distribution and open-set recognition tasks.

**Questions:**

- As shown in Table.1, the performance gap of adding maximum separation layer over baseline enlarges on more imbalanced data. However, the phenomenon does not hold in Table. 2. More explanations are needed.
- Why the proposed methods can benefit OOD detection and OSR recognition? More explanations and evidences are needed.


**Limitations:**

Yes.

**Strengths And Weaknesses:**

Strengths
- The topic of maximum class separation is interesting.
- The experimental evaluation is adequate.
- The paper is well written and easy to follow.

Weaknesses
- Limited novelty and contribution. To my knowledge, the maximum class separation has been discussed by a few works [1-2]. Moreover, the proposed method is similar with [1], and the matrix estimation in [1] does not need recursive calculation.
- Lack of insights from the experiments. The experiments mainly focus on the performance evaluation.

[1] Do We Really Need a Learnable Classifier at the End of Deep Neural Network.

[2] Dissecting Supervised Contrastive Learning.

---

> ### Author Response · Authors · 2022-08-02
> **Response to Reviewer BhPp**
>
> We thank the reviewer for their positive words on the topic, the experiments, and the writing. Below, we have addressed the listed concerns.
>
> ### Relation to other works
> Maximum separation for classification has a rich history in machine learning. The concurrent work of Yang et.al. [1] also points out the strong potential of fixed and separated class embedding in deep networks. Both works provide complementary views to maximum separation: Yang et al. highlight the link with neural collapse, while we show the potential of maximum separation beyond classification and long-tailed recognition for out-of-distribution detection and open-set recognition, with a single line of code. We will also include Graf et.al. [2] with the other referenced papers on incorporating separation through optimization in Section 5.
>
> ### Results in Table 1 versus Table 2
>
> Improvements of our approach increase with higher imbalance in Table 1 because standard architectures do not account for imbalance. In Table 2, we incorporate our approach on top of methods specifically designed for long-tailed recognition. These problem-specific approaches already tackle imbalanced settings and further benefit from our maximum separation.
>
> ### Additional analyses for OOD and OSR
>
> For out-of-distribution detection, we show in Figure 5 that our approach enlarges the gap in energy score distributions for in- and out-of-distribution samples, which results in better out-of-distribution detection. Following the reviewer’s suggestion, we have performed a feature visualization to explain the performance gain with maximum separation on open-set recognition. We have added the analysis to the supplementary materials.
>
>
> [1] Do We Really Need a Learnable Classifier at the End of Deep Neural Network.
>
> [2] Dissecting Supervised Contrastive Learning.

---

> > ### Comment · Reviewer_BhPp · 2022-08-03
> > **Additional Comment**
> >
> > Thanks for responding. Comments below detailed by remaining concerns.
> >
> > Maximum Class Separation continually draws rigorous attention in machine learning [1-6], as also stated by the authors. Considering the general classification task, we need to carefully discuss the novelty and contribution of inducing the fixed classifier and a new estimation algorithm. Also, as the formulation of Definition 1 and Lemma 1 is equivalent to Simplex Equiangular Tight Frame(ETF) [3] defined in neural collapse [3-6] and the motivation to obtain the sought-for geometry (Maximum Separation) matches with [5-6], the literature of neural collapse should be considered.
> >
> >
> > Contribution 1: Fixed classifier design
> >
> > -  As far as I'm concerned, this contribution is heavily covered in [5-6], stated as
> >     -  "For example, our experiments demonstrate that one may set the feature dimension equal to the number of classes and fix the last-layer classifier to be a Simplex ETF for network training, which reduces memory cost by over 20% on ResNet18 without sacrificing the generalization performance." [5]
> >     - "We propose a new paradigm for deep neural network with the linear classifier randomly initialized as a simplex ETF and fixed during training". [6]
> > -  Lack of  theoretical analysis. For comparison, [6] theoretically indicates that the feature learning with the fixed classifier converge to the neural collapse state, even in the imbalanced case. (Theorem 1, 2)
> > -  Fixed classifier has been discussed in many scenarios: e.g., self-supervised learning [7], long-tailed learning [8]
> >
> >
> > Contribution 2: recursive algorithm
> > - It is important to clarify why we need a recursive algorithm rather than the closed form estimation(Definition 1 in [3][6]) with satisfying calculation complexity.
> >
> >
> > [1] 2018 NeurIPS, Learning towards Minimum Hyperspherical Energy
> >
> > [2] 2021 AISTATS, Learning with Hyperspherical Uniformity
> >
> > [3] 2020 PNAS, Prevalence of Neural Collapse during the terminal phase of deep learning training
> >
> > [4] 2020 arXiv, Neural collapse with unconstrained features
> >
> > [5] 2021 NeurIPS, A geometric analysis of neural collapse with unconstrained features
> >
> > [6] 2022 arXiv, Do We Really Need a Learnable Classifier at the End of Deep Neural Network
> >
> > [7] 2022 ICLR, Understanding Dimensional Collapse in Contrastive Self-Supervised Learning
> >
> > [8] 2022 CVPR, Targeted Supervised Contrastive Learning for Long-Tailed Recognition

---

> > > ### Author Response · Authors · 2022-08-08
> > > **Response to additional comment by Reviewer BhPp**
> > >
> > > We thank the reviewer for their response and answer their additional questions below.
> > >
> > >
> > > Our solution is originally motivated by the line of works on maximum separation in deep networks. The current state-of-the-art strives for separation through approximate optimization, see e.g. [25,28,29,31,33,35] of our paper and [1,2] in the refs of the reviewer. We bring a new perspective to these works by showing that separation is not an optimization problem and can be solved in a closed form recursive manner.
> > >
> > >
> > > As noted by the reviewer, our closed-form maximum separation solution relates to observations in neural collapse, specifically NC2 in [3] from the reviewer’s references. Our approach is however different both in both formulation and empirical outcomes. In neural collapse, the final matrix is $(k+1)$ x $d$ with $d >= k+1$ for $k+1$ classes ([3,4,5] in refs above), while our matrix is more compact at size $(k+1)$ x $k$. As a result, we have a different solution in the form of a recursive algorithm. This recursive closed-form solution comes with benefits, notably it opens the door towards continual and class-incremental learning with maximum separation. The algorithm yields the maximum separation matrix near instantly for a fixed number of classes and only needs to be computed once prior to training, so does not affect the overall training and inference time. We furthermore plug our approach on top of any network architecture, rather than replace the final layer as done in e.g. [5]. Our setup focuses on improving results and we provide strong performance gains in classification and long-tailed recognition without any extra learnable parameters. We also show that maximum separation is directly beneficial for out-of-distribution detection and open-set recognition, which highlights its generic and broad applicability. We will include the references and discussion on neural collapse to Section 2 of the paper.
> > >
> > >
> > > Our theory is encapsulated in Lemma 1 and Theorem 1 and we have additional empirical analyses to help understand our approach. We show in Figure 5 that maximum separation improves out-of-distribution detection by increasing the gap in energy scores between in- and out-of-distribution samples. We have added a visualization of the outputs of closed- and open-set samples, which shows that samples from outside the closed training set are less discriminative and have a lower confidence. We have furthermore added an analysis on the Angular Fisher Score, which highlights that maximum separation increases the discriminativeness of standard networks. These new analyses have been included in the supplementary materials.

---

### Official Review · Reviewer_xgNV · 2022-07-10

**Rating:** 8
**Confidence:** 4
**Soundness:** 4 excellent
**Presentation:** 4 excellent
**Contribution:** 4 excellent

**Summary:**

The paper proposes to integrate a closed-form structural prior in deep learning architectures, which has a clear geometric interpretation. For any neural network designed for classification, hence having a final softmax layer, the contribution takes the form of a fixed (non-learnable) matrix multiplication before the computation of the softmax. For a classification problem with K classes, the construction of the matrix consists of deriving K vectors on a (K-1)-dimensional hypersphere that are maximally separated. The authors provide a simple recursive procedure to construct such a matrix that only needs to be called once before training. The proposed contribution is evaluated on different learning tasks, showcasing its relevance in various common benchmarks.

**Questions:**

Since the authors experimented with various learning tasks to showcase the relevance of their contribution, it could be of interest to additional information on the limitation of the proposed contribution.
Can the authors provide examples showcasing such limits ?

**Limitations:**

The authors stated that their work is limited to supervised settings, and cannot be applied to settings that require relational information between classes. However, in the case of unsupervised learning with variational-auto-encoders, could such a matrix be integrated at the top of the encoder for better latent space covering ?
Answering such a question could additionally broaden the significance of the contribution, since performing clustering in the latent space of an auto-encoder can provide additional information that can be very valuable in a few-shot learning setting.
Generally speaking, even if the construction of the matrix is somehow tied to a classification problem with $K$ classes, the fact that the matrix impose a geometric structure in the resulting output space should also be beneficial for unsupervised learning.

**Strengths And Weaknesses:**

The main strength of the paper is to provide a simple geometric interpretation of maximum class separation within the context of neural architecture design. Contrary to many engineering tricks used in deep learning to stabilize training or improve performances, the presented contribution does not contain any hyperparameter that needs to be fine-tuned. The originality and quality of the proposed contribution lies in the fact that the authors addressed the question of maximum separation from a theoretical point of view, and proves that their matrix design is relevant when followed by a softmax activation function. The recursive procedure used to construct the matrix closely resembles an algorithm that could be used to build an orthonormal basis for a vector space, except that in the presented case, the class vectors are required to have a fixed pairwise scalar product (equal to $-\frac{1}{K}$) instead of being orthogonal to each others. The experiments are well-chosen to prove the significance of the contribution across different challenging learning tasks, and authors focus on proving that their contribution improves on many baselines, rather than seeking for a single state-of-the-art metric in a very specific setting. Additionally, the authors shows that making their matrix learnable, either with their initialization or with random initialization, can even degrade performances, thus showing the optimality of their contribution to a certain extent. Such paper shows that it is possible to make consistent improvements in neural architectures when the chosen operations are motivated by strong theoretical arguments, somehow deviating from the traditional viewpoint that neural networks are differentiable hence optimizable black box functions. One criticism that could be made is that settings where the proposed contribution falls short are only mentioned in one sentence at the conclusion.

---

> ### Author Response · Authors · 2022-08-02
> **Response to Reviewer xgNV**
>
> We thank the reviewer for highlighting the theoretical motivation and the geometric interpretation of our work and for pointing out the need for further discussion on the limits and broader potential of maximum separation.
>
> ### Further discussion on limits of maximum separation
>
> Our approach is currently focused on multi-class settings only, where exactly one label needs to be assigned to each example. Maximum separation does also not naturally generalize to zero-shot settings, as such settings require that classes are represented by semantic vectors that point in similar directions based on semantic similarities. We position all classes equally far away from each other, hampering zero-shot generalization.
>
> ### Broader potential for unsupervised learning
>
> While we focus on the supervised setting, we agree that maximum separation has potential for unsupervised learning as well. Improving the latent space of variational auto-encoders with maximum separation sounds like an intriguing direction. We furthermore see potential in the self-supervised learning paradigm, by enforcing a maximum separation (i.e. full uniformity) between unlabelled samples in a batch.
>
> We have updated the conclusions with the above discussions.

---

### Official Review · Reviewer_9rpc · 2022-07-14

**Rating:** 7
**Confidence:** 3
**Soundness:** 4 excellent
**Presentation:** 3 good
**Contribution:** 3 good

**Summary:**

The paper proposes to bake maximum separation between classes as an inductive prior to a deep learning model. The proposed approach maximally separates class vectors, which is cleverly proven to have a closed form (assuming hyperspherical uniformity), and is implementable as a simple multiplication with a fixed matrix. Results are good on conventional tasks (Cifar, ImageNet), alongside long-tailed and open set problems.

**Questions:**

Can this applied to other tasks, like segmentation?
Can an approach like this be modified to work well in an unsupervised setting?

**Limitations:**

I'd be interested in knowing if this work can be applied to other different tasks, like segmentation, or be applied in other settings, like unsupervised.
There are obviously improvements with this method, but when I quickly ran the code a couple of times, there seemed to be a large amount of variance of the test performance, exp in the imbalanced setting. Calculating some error bars/confidence intervals would be helpful.

**Strengths And Weaknesses:**

Strengths:
The method improves classification in the imbalanced, long tailed setting, alongside OOD and open set settings.
The approach easily slots in with what appears to be any/most existing approach, as seen by the comprehensive experiments with various different methods on various problems.
Good theoretical analysis. It appears correct to the best of my understanding.
Utilizing a fixed matrix that you do not learn and improving results in a deep learning framework is an interesting approach with some novelty/originality to it.
The result of the ablation with learnable prototypes is interesting, highlighting the importance of class separation as a solid inductive prior.
How to run the code is very clearly laid out in README.md (albeit with minor mistakes).
I was able to reproduce similar numbers to a subset of the results (table 1, alexnet) using the code with minor fixes. Did not have time to try more.

Weakness:
Diagrams are confusing and could be better labeled. For example, in Figure 1, it would be most useful to label what is $P_k$ and what is $p_k$.  Figure 3 could be much more clear (sub-title each subplot with amount of imbalance).
Some figures appear to contain the wrong description i.e. figure 4 "green" (there is no green).
The code was troublesome to get running; many simple errors to fix.

---

> ### Author Response · Authors · 2022-08-02
> **Response to Reviewer 9rpc**
>
> We thank the reviewer for their positive comments regarding the novelty of the approach, the theoretical analysis, and the empirical effectiveness about our paper and also for their guidance to improve the paper and code.
>
> ### Figures and code
>
> We thank the reviewer for the suggestions to improve Figure 1 and Figure 3. We have added $Pk/pk$ to Figure 1 and added plot labels with imbalance ratios in Figure 3. The updated figures are shown in the revised pdf. We have also fixed the typo in the caption of Figure 4. We will address the minor code fixes and upload it to a public github repository.
>
>
>
> ### Generalizing to unsupervised settings
>
> Our paper focuses on the supervised setting, but we also see potential for maximum separation in unsupervised settings. For example, Wang and Isola (ICML, 2020) have previously shown that self-supervised learning involves optimizing for alignment and uniformity. Maximum separation can potentially improve self-supervised learning by increasing uniformity between samples in batches for contrastive learning. Also segmentation, which generalizes classification to the pixel-level, is a potentially fruitful direction for maximum separation. We have added this discussion to Section 5.
>
> ### Error bars for imbalanced experiment
> Based on the reviewer’s suggestion, we have run the experiments in Table 1 of the main paper 5 times and added error bars. The results show that over multiple runs, the improvements are stable. Due to space limitations, we have added the experiment with error bars to the supplementary materials.
>
>
>
>
>
>
> | |  | | CIFAR-10 |  |  |  |  | CIFAR-100 |  | |
> |--- |--- | --- | --- | --- | --- | --- | --- | --- | --- | --- |
> | | -            | 0.2           | 0.1          | 0.02 | 0.01 | - | 0.2 | 0.1 | 0.02 | 0.01 |
> | ConvNet       | 56.45± 0.32   | 45.88 ± 0.43 | 40.04± 0.38 | 27.17± 0.52 | 16.31 ± 0.22 | 86.30± 0.21 | 78.37± 1.04 | 73.6± 0.58 | 51.71 ± 0.38 | 42.72 ± 1.21 |
> | \+ This Paper | **57.05± 0.55**   | **46.21± 0.45**  | **40.44± 0.23** | **28.16 ± 0.31** | **18.15 ± 0.53** | **86.48± 0.20** | **79.44± 1.20** | **75.4 ± 1.03** | **56.98 ± 1.16** | **48.26 ± 0.65** |
> | ResNet-32| 75.42 ± 0.37 | 65.20± 0.43  | 58.01± 1.01 | 42.70± 0.20 | 34.98± 0.54 | 94.41±0.25 | 87.96± 0.24 | 82.95± 0.45 | 68.04± 0.83 | 56.5± 0.56 |
> | \+ This Paper | **76.41± 0.21**   | **66.22±0.56** | **60.23± 0.54** | **45.11± 0.13** | **37.65± 0.81** | **96.12± 0.19** | **91.26± 0.22** | **88.01± 0.73** | **77.12± 1.33** | **68.8± 1.42** |

---

### Official Review · Reviewer_bBZR · 2022-07-16

**Rating:** 7
**Confidence:** 4
**Soundness:** 4 excellent
**Presentation:** 3 good
**Contribution:** 3 good

**Summary:**

This paper proposes a closed-form design of the classifier layer, which encourages the maximal separation between any two classifiers on a hypersphere. The design is simple and straighforward, and it introduces a recursive algorithm to place $k+1$ classifiers in $k$ dimensional space. The experimental results demonstrate effectiveness in standard visual recognitio, long-tailed recognitio, OOD detection and open-set recognition.

**Questions:**

- My major questions are given in the "Strengths And Weaknesses" section.

- Some visualization on the learned features could really improve the paper. I am also curious about how the learned features will be given a fixed set of classifiers.

- While the classifiers are no longer learned, I am wondering whether the convergence performance will also be improved (since less number of parameters are needed to learn)? Some experiments on convergence speed (eg, iteration vs. testing accuracy) would be nice.

**Limitations:**

For technical limitations, I have discussed them in the "Strengths And Weaknesses" section. For potential negative societal impact, I am unaware of any.

**Strengths And Weaknesses:**

Strengths:

- This paper is well written and structured. I generally enjoy reading this paper and find this idea quite interesting. Placing static classifiers and encourage their maximal separability is very natural and well motivated. The inductive bias from hyperspherical uniformity is encoded in the classifier layer which is used to generate corresponding gradients to update the whole network. Due to such a design, I think the learned features should also be quite discriminative (maybe with some sort of angular margin). I would suggest the authors to conduct some visualizations or Fisher discrimination analysis (say angular Fisher score in Appendix E of [1]) to demonstrate such advantages in a more intuitive way.

- The resursive algorithm is neat and efficient. I like its simplicity and the theoretical analysis that motivates it. The analysis is intuitive and should be correct as far as I'm concerned. I am wondering how it compares to gradient-based optimization. For example, you can directly minimize a matrix's hyperspherical energy to obtain its maximally separable column vectors, and then keep them fixed during the network training. Although it may not be the optimal design, I am curious how it compares empirically to the proposed method.

- I believe the static design of the classifier layer is of sufficient interest to the community and is of sufficient novelty, because it may be a potential solution to both encode better inductive bias and address the computational difficulty of training million-level number of classes. Besides the proposed static design, I would point the authors to a probabilistic design (in Section 6.4 of [2]) that also pre-specifies a set of fixed classifiers (which are randomly initialized) and then learns an orthogonal matrix that applies to them. I find the same idea can also be used in this paper, meaning that you learn an orthogonal matrix (using the same way as [2]) just to rotate/reflect these maximally separable classifiers. This does not change their pairwise distance / similarity, which means their hyperspherical energy stays the same. I think it will further improve the proposed method by granting it more flexibilty (while their maximal separability does not change).

- The paper demonstrates the effectiveness of the hyperspherical uniformity inductive bias in a number of recognition experiments. Its generalization ability is well verified. To me, the experiments should also be easily reproducible.


Weaknesses:

- While I like the general idea of static classifier design, the proposed method has an obvious weakness: the dimensionality of classifier scales linearly with the number of classifiers. This may be okay for benchmarks like CIFAR and ImageNet, but it still constrains its practical usage in large categorical training (say extreme classification). I can understand this particular algorithm design (ie, $k+1$ classifiers in $k$-dimensional space) is due to the fact that static assignment of maximally separable classifiers is very challenging when the dimensioanltiy is way smaller than the number of classifiers. I think this limitation, as a future direction to study, should be explicitly discussed in the paper.

- I appreciate the simplicity of the static classifier design, but the lack of flexibility (learnability) may limit its performance. Learning an orthogonal transformation (or even simpler, learning a rotation matrix) can be beneficial, as I mention in the third point in Strengths.

- As Lemma 1 shows, the inner product approaches zero as $k$ becomes larger. Does it mean that we can use orthogonal matrix (which is $k\times k$ instead of $k+1 \times k$) to replace the maximally separated matrix? It seems to be an interesting conclusion that one can take advantage of. This is not exactly a weakness, but it could be an interesring connection to orthogonality. Maybe the authors can comment / elaborate on this.

- The definition of maximally seprated matrix is a bit inappropriate in the sense that the equal inner product between any two classifiers does not always hold for matrices of any size. It only holds when the matrix is of the size $(k+1)\times k$. I suggest the authors to clarify this in order to avoid confusion.


Summary:
- I am in favor of the core idea (as well as the overall direction in static classifier design) and find this paper a solid work in general. I would vote for clear acceptance given the authors properly address my concerns.


[1] SphereFace: Deep Hypersphere Embedding for Face Recognition, CVPR 2017

[2] Orthogonal Over-Parameterized Training, CVPR 2021

---

> ### Author Response · Authors · 2022-08-02
> **Response to reviewer bBZR**
>
> We thank the reviewer for their detailed review and thoughtful comments. We discuss the additional analyses and questions below.
>
> ### Angular Fisher score analysis
>
> We reported the Angular Fisher Score from Liu et.al. [1] in the table below for CIFAR-10 and CIFAR-100 test sets. We trained a ResNet-32 with the same settings as Table-1 from the paper. For the Angular Fisher Score, lower is better. Across datasets and imbalance factors, the score is lower with maximum separation, providing additional verification of our approach. We have added the angular Fisher score analysis to the supplementary materials.
>
>
> ||  | CIFAR-100 |  |  | CIFAR-10 |  |
> | ---        |---        |---        |---        |---        | ---        | ---         |
> || - | 0.1 | 0.01 | - | 0.1 | 0.01 |
> | ResNet-32      | 0.2954    | 0.4958 | 0.7272 | 0.058  | 0.2305 | 0.4141
> | + This Paper | **0.1521** | **0.4483** | **0.6952** | **0.055** | **0.1397**  | **0.3240** |
>
>
>
> ### Comparison to optimization-based separation
> Following the reviewer’s suggestion, we have compared our approach to a baseline that optimizes for class vectors through optimization and fixes the vectors afterwards. We compare to the hyperspherical prototype approach of Mettes et al. [3]. We have looked into the class vectors themselves, as well as the downstream performance. For the class vectors, we find that a gradient-based solution has a pair-wise angular variance of over one degree for 100 classes, indicating that not all classes are equally well separated, while we do not have such variability. We have also performed additional long-tailed recognition experiments for our maximum separation approach versus the hyperspherical prototype approach of Mettes et al [3] with ResNet-32 backbone. Below are the results for CIFAR-10 and CIFAR-100 for three imbalance ratios:
>
>
>
> || | CIFAR-100 | | | CIFAR-10 | |
> | --- | --- | --- | --- | --- | --- | --- |
> | | - | 0.1 | 0.01 | - | 0.1 | 0.01 |
> Mettes et.al. | 71.58 | 53.28| 34.08 | 93.27 | 86.16 | 61.63 |
> This Paper | **76.23** | **60.54** | **38.85** | **95.09** | **88.16** | **69.70** |
>
> We conclude that a closed-form maximum separation is preferred for recognition. We have added the comparison to the supplementary materials.
>
> ### Relation to orthogonality
>
> We agree that the inner product approaches zero and angles are closer to being orthogonal as the number of classes $k+1$ becomes larger. Substituting it with an $(k+1)$x$(k+1)$ orthogonal matrix (say, an orthogonal basis) still uses only the positive subspace of the hypersphere and is hence not maximally separated. Perhaps a better choice of orthogonal matrix as indicated by the reviewer from [2] might be useful to get performances similar to our maximum separation. Similarly, learning an orthogonal rotation/reflection from the probabilistic classifiers from [2] would also be an interesting connection to maximum separation for future research.
>
>
> ### Number of dimensions
>
> Indeed, the output dimensionality is always one less than the number of classes, hence it scales linearly to many classes as well. This is the same for standard cross-entropy, where the number of output dimensions needs to be the same as the number of classes, hence our approach does not provide any burdens on top of standard softmax cross-entropy optimization. We have furthermore clarified in line 83 that the maximum separation only holds for matrices of $k$x$(k+1)$ for $k+1$ classes. Lastly, we observe that the down-stream recognition convergence is similar with and without maximum separation.
>
> [1] SphereFace: Deep Hypersphere Embedding for Face Recognition, CVPR 2017
>
> [2] Orthogonal Over-Parameterized Training, CVPR 2021
>
> [3] Hyperspherical Prototype Networks, NeurIPS 2019

---

> > ### Comment · Reviewer_bBZR · 2022-08-06
> > **Thanks for the response**
> >
> > Thanks for the response. I appreciate the effort put in the rebuttal, and I am satisfied with the response. However, while I am generally satisfied with the response, I would like to clarify one misunderstanding from the authors. But I do want to note that although this is a limitaiton for this work, it does not prevent this paper to be an interesting and solid idea (which may inspire others to work on solving this limitation).
> >
> >
> > "Indeed, the output dimensionality is always one less than the number of classes, hence it scales linearly to many classes as well. This is the same for standard cross-entropy, where the number of output dimensions needs to be the same as the number of classes, hence our approach does not provide any burdens on top of standard softmax cross-entropy optimization."
> >
> > - I think you may misunderstand my point. My point is that when the static design of classifiers requires the dimension of classifier (which is the same as the dimension of feature) to scale linearly with the number of classes. This implies that the feature will have really large dimensionality if the number of classes is large (say million-scale classification), while this is not the case for standard softmax cross-entropy. Usually for million-scale classification, you could still set the feature dimension as 512 or 1024 for standard learnable classifiers, which can be way smaller than the number of classes.

---

> > > ### Author Response · Authors · 2022-08-08
> > > **Response to additional comment by Reviewer bBZR**
> > >
> > > We thank the reviewer for their response.
> > >
> > > Regarding feature dimensionality, you can indeed set feature embedding to eg. $512$ dimensions with a standard softmax cross-entropy formulation. To obtain class logits and compute the loss, a final layer with $512$ x $(k+1)$ learnable parameters is then needed for $k+1$ classes (excluding the bias). Similarly in our approach, we can set the number of feature dimensions to $512$. This is then followed by a learnable layer of size $512$ x $k$. On top, we add our fixed matrix of size $k$ x $k+1$ to get class logits. This means that in terms of learnable parameters, there is no noticeable difference between the standard setup and our maximum separation formulation. With many classes however, the fixed final matrix will be large, which can lead to extra computational effort. At the ImageNet scale (1,000 classes) training and inference times are similar, but we have yet to investigate extreme classification cases. We hope this helps clear up the similarities and differences, we will add this point to the paper.

---

> > > > ### Comment · Reviewer_bBZR · 2022-08-08
> > > > **Thanks for the response**
> > > >
> > > > Thanks for the response. I have no more questions, and I hope these suggestions can be useful when preparing a revised version.

---

### Meta-Review · Area_Chair_tn9R · 2022-08-26

**Recommendation:** Accept
**Confidence:** Certain

**Metareview:**

This paper aims at introducing a criterium for class separation. The paper demonstrates high performance, by proposing an affine transformation of the canonical embedding of labels, which lead to a maximal separation between those new vectors.
Given the simplicity and good numerical results, I recommend accepting this paper; however, the minor revisions suggested by reviewer BhPp needs to be addressed in the camera-ready version.

**Award:**

No

---

### Decision · Program_Chairs · 2022-09-14

Accept

---

> ### Public Comment · ~Federico_Pernici1 · 2022-11-30
> **Regular Polytopes and d-simplex fixed classifier...**
>
> This paper substantially proposes with a different name the same solution proposed in the paper "Regular Polytope Networks", published in Feb. 2021 in IEEE Transaction on Neural Networks and Learning Systems [1], where theoretical justifications of maximal separation are also provided in the general framework of Regular Polytopes. In fact, the fixed matrix in this manuscript is basically the same d-simplex fixed classifier presented in [1]. Surprisingly, the related citations are missing and the Reviewers/ACs didn't help in this regard which is, in our opinion, somewhat critical for an oral presentation at NeurIPS. Furthermore, the original concept appeared in [2] in 2019 and in [3] in 2020.
>
> References
>
> [1] Pernici, Federico, Matteo Bruni, Claudio Baecchi, and Alberto Del Bimbo. "Regular polytope networks." IEEE Transactions on Neural Networks and Learning Systems (TNNLS) Feb. 2021.
>
> [2] Pernici, F., Bruni, M., Baecchi, C. and Del Bimbo, A., 2019, January. Maximally Compact and Separated Features with Regular Polytope Networks. In CVPR Workshops (pp. 46-53).
>
> [3] Pernici, Federico, Matteo Bruni, Claudio Baecchi, Francesco Turchini, and Alberto Del Bimbo. "Class-incremental learning with pre-allocated fixed classifiers." In 2020 25th International Conference on Pattern Recognition (ICPR), pp. 6259-6266. IEEE, 2020.

---

> > ### Public Comment · Authors · 2022-12-06
> > **Reply to Regular Polytopes and d-simplex fixed classifier...**
> >
> > Dear Federico,
> >
> > Thank you for pointing out the related work on regular polytope networks. The notion of learning with regular simplex has come back a number of times in literature and we agree that your paper should have been included in this discussion. We will write a follow-up version and will include the provided references. Overall, our paper provides a different formulation and additional theory what constitutes optimality. The paper is also highly empirical, showing the potential of learning with this approach for long-tailed recognition, out-of-distribution detection, and open set recognition.
> >
> > Kind regards,
> >
> > NeurIPS #1648 authors